# Remediation of Sb-Contaminated Soil by Low Molecular Weight Organic Acids Washing: Efficiencies and Mechanisms

Sicheng Li [1], Weibin Pan [1,*], Lizhi Tong [2], Yuanyuan Hu [1], Yulin Zou [1] and Xiaojia Huang [1]

1   School of Environment and Energy, South China University of Technology, Guangzhou 510006, China
2   South China Institute of Environmental Sciences, Ministry of Ecology and Environment, 7 West 12 Street, Guangzhou 510655, China
*   Correspondence: ppwbpan@scut.edu.cn

**Abstract:** Low molecular weight organic acids (LMWOAs) are promising agents in the remediation of heavy metal contaminated soil with strong complexing ability and less environmental impact. However, the application of LMWOAs for washing the Sb-contaminated soil still faces great challenges, such as the selection of suitable washing agents, optimal washing parameters, and the unclear Sb removal mechanism. In this study, five suitable LMWOAs were screened from ten common washing agents and their optimum washing parameters were determined. The results showed that oxalic acid (OA) and HEDP were the top two outstanding agents, and the removal efficiencies of Sb were 68.79% and 49.73%, respectively, under optimal parameters (OA at 0.5 mol/L, HEDP at 0.2 mol/L, washing for 480 min, and the liquid-to-solid ratio of 15). The soil was analyzed for chemical speciation, morphology, functional groups, and mineralogy before and after washing. The results indicated that Fe/Al minerals in the soil are the main reason for the adsorption of Sb, and the possible mechanisms of Sb removal by LMWOAs included the dissolution of minerals, complexation reaction, and ligand exchange. Our findings highlight the potential application of LMWOAs as efficient washing agents to remove Sb from contaminated soils.

**Keywords:** soil washing; antimony; organic acid; oxalic acid; HEDP; desorption





## 1. Introduction

Antimony (Sb) is a metalloid element in Group VA of the periodic table, as well as arsenic (As). Sb generally occurs as oxides, hydroxides, or oxyanions either in the +5 state in oxic environments or the +3 state in anoxic environments [1]. China has the most plentiful Sb resources of any country in the world. In 2019, China produced approximately 60% of the world's Sb [2]. However, as a result of antimony mining, smelting, and usage of Sb-containing industrial products, large amounts of Sb are inevitably released into the soil environment, posing an ecological risk [3–5]. For instance, the Sb concentration in the soil around Xikuangshan mine in Lengshuijiang, China, has reached $101 \sim 5045 \text{ mg} \cdot \text{kg}^{-1}$, which far exceeds the maximum permissible concentration of $36 \text{ mg} \cdot \text{kg}^{-1}$ for Sb in soil stipulated by the World Health Organization (WHO) [6]. Sb and its compounds are harmful to human health and even carcinogenic [7]. In addition, Sb has been designated a priority pollutant by the European Union and the United States [8]. Sb is already a pollutant of emerging concern [2,9]. Adopting practical and effective remediation technology is therefore, urgently needed to remove Sb from contaminated soil.

Currently, most studies on remediation methods for Sb-contaminated soil include solidification/stabilization, phytoremediation, and soil washing [2,10,11]. Stabilization technology is the most researched remediation method for Sb-contaminated soil. Iron oxides and oxyhydroxides exhibit promising properties when employed to stabilize Sb-contaminated soils [12]. Moreover, biological amendments, such as poultry compost and sheep compost, help to reduce the soluble and exchangeable fraction of Sb in soil that

is available to plants [13]. Nevertheless, the long-term stabilization effect is unknown, and there is still a risk of leaching of heavy metal ions, even the overall stability of the soil environment may be impacted [14]. Phytoremediation is a feasible green remediation technique for Sb-contaminated soil, but its long cycle time also limits its application [2,15]. Soil washing, as the name suggests, utilizes a washing agent to fully contact heavy metals and then remove them from the soil [16]. In comparison with other remediation approaches, soil washing is an efficient way of treating metal-polluted soil since it can permanently and swiftly extract metals from the soil [17,18]. Serafimovska et al. [19] applied a four-step sequential extraction procedure (BCR) to analyze the species composition of Sb in soil contaminated from the coal-fired power plant, discovering 79~97% residual state content of Sb. Therefore, it is not easy to remove Sb from the soil. Navarro et al. [20] used seawater to flush severely metal-contaminated sites by metal smelting slags created by metallurgical activities in the Sierra Almagrera and discovered that immobilization occurred because of the presence of ferrihydrite, resulting in the removal of only negligible amounts of Sb. Tokunaga et al. [21] investigated the extraction behavior of soil pollutants and their components using mineral acids ($HNO_3$, $HCl$, $H_3PO_4$, and $H_2SO_4$), organic acid salts (citrate and tartrate), and artificial chelating agents (EDTA and DTPA) as extracting agents, finding that the extraction rate of $H_2SO_4$ and $H_3PO_4$ for Sb were higher than other extractants, which were less than 20%. Guemiza et al. [22] investigated countercurrent leaching methods with $H_2SO_4$ + $NaCl$, after eight cycles, the average Sb removal yield was $51.1 \pm 4.8\%$. Sun et al. [23] applied tartaric and malic acid to wash Sb-contaminated soil and achieved 23.21% and 21.88% removal of Sb, respectively. To sum up, there are two technical issues with the use of soil washing methods to remove Sb from soils: 1. The removal efficiency of previous washing reagents applied to Sb-contaminated soil is generally low, 2. Harmful reagents such as inorganic strong acids or too many washing cycles were applied, which not only severely damaged the soil structure, but also made it difficult to apply to practical remediation. Soil washing is a very intricate process, which is reflected in both its distinction as a phenomenon and mechanism [24]. The factors and conditions affecting its performance are still unclear, particularly in the case of the Sb-contaminated soil. However, compared to metals such as Cd, Pb, or, As, there have been insufficient investigations on the technology of soil washing to remediate Sb-contaminated soil [2,25,26], limiting the application of this technology in Sb-contaminated soils.

It should be noted that specific washing agents play a significant role in determining washing efficiency [25]. In heavy metal-contaminated soil, two basic types of washing agents have received a great deal of attention and use: Inorganic strong acid and Chelating agents [27,28]. However, the direct use of inorganic strong acid solution during soil remediation can severely impact the physicochemical properties and organic matter in soil [29]. Although artificial chelating agents represented by ethylenediaminetetraacetic acid (EDTA) have good heavy metal removal effects [30,31], the EDTA remaining in the soil after washing is hard to be biodegraded, causing secondary pollution [32]. LMWOAs are excellent examples of washing agents due to their outstanding effect, ease of access, and strong biodegradability [33], which ensures that great remediation results are achieved with less disturbance to the soil environment. Through bridging, complexation, adsorption site competition, reduction, and dissolution, LMWOAs had an impact on the migration and transformation of heavy metals in the soil [34,35]. Currently, the effectiveness of LMWOAs in the washing of cationic metals (e.g., Cd, Cu, Zn, etc.) contaminated soils has been confirmed [36,37], but the applicability to polluted soils with oxygenated anionic metals like Sb has not yet been adequately investigated. It should be mentioned that the remediation effects of the same agent on different heavy metal contaminated soils may be very different because Sb, which tends to exist as oxygen anions, present the opposite charge to other cationic heavy metals (e.g., Cd and Pb) [1,38]. Consequently, it is necessary to discover effective LMWOAs as washing agents and to investigate their optimization conditions, heavy metal removal effectiveness, distribution for soil washing, and mechanisms. In addition to washing agents, many factors can influence the effectiveness of soil washing, such

as agents 'concentration, washing time, liquid/soil ratio (L/S ratio), etc. [39]. Optimization of washing parameters is also an important way to improve Sb removal efficiency.

Therefore, the objective of this study is (1) the selection of the optimal LMWOAs for the Sb-contaminated soil washing, (2) the determination of their optimal washing parameters, and (3) the exploration of the effects and mechanisms of washing on the soil. Three influencing parameters, including agent concentration, L/S ratio, and washing time, are considered based on the findings of earlier theoretical and experimental studies on soil washing. In addition, soil morphology, functional groups of soil, soil minerals, and speciation of Sb were analyzed before and after washing, expecting to explore the washing mechanism initially. The LMWOAs screened in this study were efficient and green, and one of them was applied for the first time to Sb-contaminated soil. This study also revealed the primary mechanism of Sb removal from soil by washing. This work can provide practical information and technical support for actual soil-washing technology for the remediation of Sb-contaminated soil.

## 2. Materials and Methods

### 2.1. Soil Sample Preparation

The uncontaminated soil was taken from the 0–20 cm soil layer in a park in Guangzhou, Guangdong Province, afterward, the soil was transported to the laboratory, air-dried, and picked away obvious debris such as stones and plant roots, then ground to pass through a 2 mm sieve. To prepare Sb-contaminated soil samples with Sb content of 400 mg/kg, a certain concentration of Potassium pyroantimonate ($K_2H_2Sb_2O_7$) solution was added into the 5-kg dry mass of uncontaminated soil to achieve about 40% moisture content. To guarantee the soil was homogenized, the soil used for tests was mixed well. The contaminated soils were placed into a polyethylene plastic box and maintained the moisture content of the soil at a consistent level during the 30-day aging period at room temperature. After that, the contaminated soil samples were naturally dried and filtered through a 2 mm sieve in preparation for the following experiments. The $K_2H_2Sb_2O_7$ was of AR grade and was acquired from Shanghai Macklin Biochemical Co., Ltd. (Shanghai, China). Other reagents were also of AR grade.

### 2.2. Washing Experiments

The Soil washing experiments were carried out in 50 mL centrifuge tubes and repeated three times. For each experiment, 1.00 g of soil was washed with 5–30 mL of washing agents. The tubes were shaken horizontally at 250 rpm at 25 °C in a shaking bed, after that, the mixture was centrifuged (4000 rpm, 10 min) and filtered through a 0.45 μm pore size membrane for the filtrate Sb concentration. First, ten commonly used washing agents were preliminarily screened, and the preliminary screening parameters were: Agent concentration 0.1 mol/L, liquid-soil ratio (L/S ratio) 10:1, and washing time 360 min. The ten agents were succinic acid (SA), acetic acid (HAc), tartaric acid (TA), malic acid (MA), citric acid (CA), etidronic acid (HEDP, organic phosphonic acid), oxalic acid (OA), glutamic acid diacetic acid, tetrasodium salt (GLDA), EDTA, and $KH_2PO_4$. Based on the Sb removal efficiency, five excellent washing agents, which were OA, HEDP, CA, MA, and TA, were finally chosen. Then the five washing agents were used for parametric studies to investigate the effect of agent concentration, L/S ratio, and washing time on Sb removal effectiveness. The following operational parameters adjustments were made specifically: Concentration (0.01, 0.02, 0.05, 0.1, 0.2, 0.5, 0,7, 1.0 mol/L), L/S ratio (5/1, 10/1, 15/1, 20/1, 30/1), washing time (10, 30, 60, 120, 240, 360, 480, 600, 720 min). The above gradients of the parameters and other conditions of the washing experiments were determined by summarizing the earlier research on the soil washing technique for heavy-metal-contaminated soils [23,24,37,40,41] to obtain the optimum performance of the washing agents.

### 2.3. Sequential Extraction of Sb from Soil

Sequential chemical extraction of soil was performed using a modified method proposed by Okkenhaug et al. [42]. This method was originally developed for As based on extractants typically employed for the oxyanions phosphorous (P) and selenium (Se) [43]. Because Sb also desorbs from crystalline Fe oxides in the presence of oxalate, the ammonium-oxalate buffer extraction that was used originally to remove amorphous Fe, Al, and Mn-oxides was not employed in the modified method [42]. The samples were fractionated into four major forms: Non-specifically adsorbed, specifically adsorbed, amorphous and crystalline Fe and Al oxides bound, and the residual fraction. All the details of the modified extraction procedure in this research are shown in Table 1.

**Table 1.** Modified Wenzel extraction procedure.

| Fraction | Extractant | Extraction Conditions |
|---|---|---|
| Non-specifically adsorbed fraction (F1) | $0.05 \text{ mol·L}^{-1}$ $(NH_4)_2SO_4$ | 4 h shaking, 25 °C, L/S ratio 25:1 |
| Specifically adsorbed fraction (F2) | $0.05 \text{ mol·L}^{-1}$ $(NH_4)H_2SO_4$ | 16 h shaking, 25 °C, L/S ratio 25:1 |
| Amorphous and crystalline Fe and Al oxides bound fraction (F3) | $0.2 \text{ mol·L}^{-1}$ $NH_4$-oxalate buffer and $0.1 \text{ mol·L}^{-1}$ ascorbic acid, pH = 3.25 | 0.5 h in a water basin at $96 \pm 3$ °C in the light, L/S ratio 25:1 |
| The residual fraction (F4) | Concentrated HCl, $HNO_3$ in a ratio 3:1 | 180 °C, 30 min |

### 2.4. Analysis and Determination Methods

The analysis and determination methods are performed concerning the corresponding standards issued by China. The soil pH was measured in a 1:2.5 soil-to-water using a pH meteST300 (OHAUS Instruments Co., Ltd., Changzhou, China) [44]. The soil total organic matter (OM) was determined by the potassium dichromate volumetric method [45]. The cation exchange capacity (CEC) was determined by the hexamminecobalt trichloride solution-spectrophotometric method [46]. Antimony content in the solution was tested using a flame atomic absorption spectrometer (FAAS, SHIMADZU AA-6880, Shimadzu, Kyoto, Japan), and the Sb content of the soil was tested after the aqua regia digestion [47]. The grain sizes of the soil were determined using a Soil particle sizer (Malvern Mastersizer 2000, Malvern Panalytical, Malvern, UK).

An X-ray fluorescence spectrometer (XRF, Bruker S6 Jaguar, Bremen, Germany) was used to analyze the main elemental composition of the soil. Scanning electron microscopy (SEM, Tescan MIRA LMS, Brno, Czech Republic) and X-ray diffraction (XRD, Rigaku Smartlab 9KW, Tokyo, Japan) were used to examine the change in the morphology and the mineral structure of the soil. The following operating conditions were used in XRD analysis: Cu-K$\alpha$ monochromatic radiation, 2°/min, 5–90°. The MDI Jade 6.5 software (Materials Data Inc., Livermore, CA, USA) was used to identify the presence of mineral phases. An Energy dispersive spectrometer (EDS, Tescan MIRA LMS) was used to analyze the elemental distribution of the soil. Fourier transform infrared (FTIR, Thermo Scientific iN10, Waltham, MA, USA) was used to detect changes in the chemical groups of soil. Soil samples for the above instrument analysis were obtained under the following conditions: 0.5 mol/L OA, 0.2 mol/L HEDP, L/S ratio 15:1, washing time 480 min, shaking speed 250 rpm, and temperature 25 °C. The washed soil was dried in a constant temperature oven at 65 °C for more than 24 h and then sieved to obtain soil samples for instrumental analysis.

### 2.5. Quality Control and Statistical Analysis

Glass and plastic vessels used in this work were presoaked in 10% $HNO_3$ overnight. Each experiment in this study was repeated three times, and the data are provided as a mean with a standard deviation. The one-way ANOVA was used to analyze for significant differences between treatments [48]. Four kinetic models, namely Elovich, pseudo-first-order, pseudo-second-order, and two-constant equation models, have been used to fit the kinetic data. The coefficients of determination ($R^2$) and standard error (SE) were computed

to obtain the best-fitting kinetic model [39,49,50]. The equation for the removal efficiency of Sb (%) was as follows:

$$Removal\ efficiency = \frac{C_{solution} \times V}{m \times C_{total}} \times 100\% \tag{1}$$

where $C_{solution}$ is the concentration of Sb in the washing solution (mg/L), $V$ is the volume of the washing solution (L), $m$ is the dry mass of soil (kg), $C_{total}$ is the total concentration of Sb in dry soil (mg/kg).

## 3. Results and Discussion

### 3.1. Properties of Contaminated Soil Samples

Table 2 presents the physical and chemical characteristics of the contaminated soil samples. Successful preparation of artificially contaminated soil samples revealed that the Sb level was 384.54 mg/kg, which was around ten times higher than the regulation limitations set for type I land in the Soil Environmental Quality-Risk Control Standard for Soil Contamination of Development Land (GB 36600-2018). The soil samples were alkaline, with a pH of 8.63. The soil samples were classified as sandy loam according to the USDA soil texture triangle. XRF analysis showed that Si (25.88%) was the most abundant element in the soil (C and O elements could not be detected), followed by Al (15.74%) and Fe (10.69%). In addition, both OM and CEC test values were very low, and the soil was significantly Fe-rich and Al-rich, which is a typical feature of soils in southern China [51].

**Table 2.** Physical and chemical properties of the contaminated soil samples.

| Parameters | Values |
|---|---|
| pH | 8.63 |
| OM/(g·kg$^{-1}$) | 3.155 |
| CEC/(cmol·kg$^{-1}$) | 2.119 |
| Total Sb/(mg·kg$^{-1}$) | 384.54 |
| Clay/% | 6.99 |
| Silt/% | 26.78 |
| Sand/% | 66.23 |
| Si/% | 25.88 |
| Al/% | 15.74 |
| Fe/% | 10.69 |
| K/% | 3.34 |

### 3.2. Washing Agents Screening

Ten commonly used soil-washing agents were screened to remove Sb from the ground. As seen in Figure 1, the amount of Sb leached from the soil by HAc was only 2.06%, which indicated that Sb was little in the water-soluble/weak acid-extracted state of the soil [52]. Similarly, the efficiency of SA was very unsatisfactory, with only 1.94%. Washing efficiencies of other different LMWOAs for Sb were OA > HEDP > CA > MA ≈ TA. Among them, OA performed the best, with a removal efficiency of 58.88%. It was noteworthy that HEDP showed the second-highest efficiency and that this reagent was applied for the first time to treat Sb-contaminated soil. Studies have shown that complexation is one of the crucial elements determining the removal of heavy metals by LMWOAs [53]. For instance, the carboxyl group (–COOH) of carboxylic acid can be complex with Sb, and thus the corresponding organic reagent could remove it [4]. Although SA and HAc also had -COOH, their removal efficiencies were much lower than those of other LMWOAs. It indicated that the Sb removal efficiencies of different LMWOAs were determined by a combination of factors.

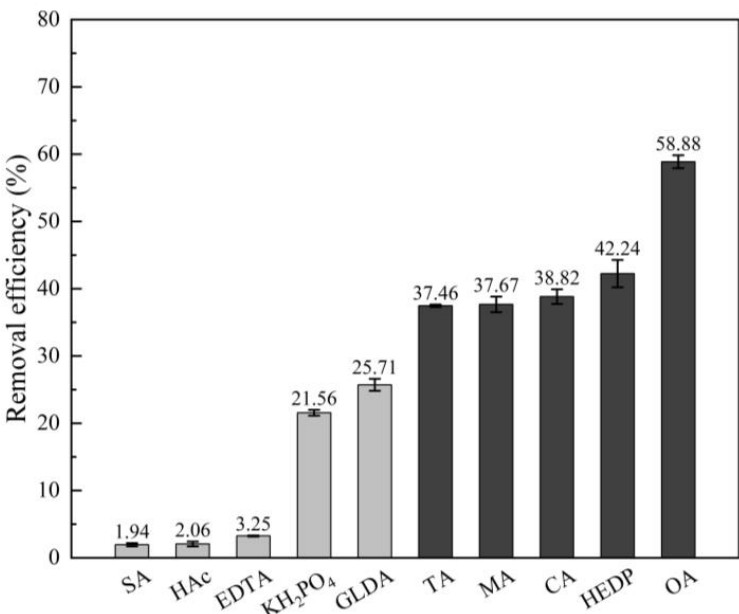

**Figure 1.** Removal efficiencies of Sb by different washing agents with the same concentration (0.1 mol/L). Other experimental conditions were: Washing time 360 min, L/S ratio 10/1, shaking speed 250 rpm, and temperature 25 °C.

Previous studies have shown that organic acids' main chemical mechanisms for dissolving metals are acidification, complexation, and reduction [54,55]. OA has the lowest pH among them (the pH of 0.1 mol/L OA was about 1.3), and a low pH environment promotes the dissolution and further complexation of heavy metals [56]. Besides, OA serves as a powerful chelating agent and a mild reducer (Eo = −18), which might result in the formation of potent complexes with the Fe released from the soil when the Fe oxides are dissolved [40]. These complexes presumably facilitated the extraction of Sb by stemming the formation of a new Fe oxide phase which prevents the re-adsorption or re-precipitation of Sb. It is worth noting that HEDP (organic phosphonic acid) achieved the second highest clearance efficiency, 42.24%, because the phosphate groups of HEDP, which have a similar structure to the antimonate ions, can compete with the antimonate ions in the soil for adsorption sites on the surface of iron oxides and then undergo ligand exchange, thus releasing Sb [23,57]. The washing efficiency of KH2PO4, which also could release $PO_4^{3-}$, was 21.56%, corroborating the above mechanism. Zeng used phosphate to leach As-contaminated soil and drew a similar conclusion [58]. Wang proposed that some of the As oxygen ions originally bound to complexing substances such as soil humus were replaced by –COOH of CA, resulting in improved removal of As [41]. Often Sb is thought to act similarly to As, not always with justification [1]. Kim used 0.1 mol/L of EDTA to extract heavy metal-contaminated soil around the smelter and accomplished an As extraction rate of 48% [59]. However, the extraction rate of Sb in this experiment was only 3.25% at the same concentration of EDTA. The washing efficiency of GLDA, which is also an artificial chelator, was 25.71%.

The removal efficiencies of Sb with different LMWOAs vary greatly, one explanation is that different organic acids have different molecular structures, which causes the generation and stability of reaction products to vary. Based on the above screening results, the top five LMWOAs were selected for the next step of the experiment. They were OA, HEDP, CA, MA, and TA, respectively.

### 3.3. Optimization of Soil Washing Parameters

3.3.1. Concentrations of Washing Agents

The concentration of the washing agent plays a crucial role in removing Sb from soils by changing the amount of substance participating in the reaction process [37,60]. Determining the optimal concentration of a single LMWOA is an essential part of determining the washing parameters, which can significantly improve the Sb removal efficiency. The removal efficiency of Sb using different concentrations of CA, OA, TA, MA, and HEDP are summarized in Figure 2A. The removal efficiency increased significantly when the concentrations of LMWOAs were increased in the range of 0.01 mol/L to 0.1 mol/L. The efficiencies of CA, OA, TA, MA, and HEDP increased from 31.48%, 29.71%, 24.75%, 26.95%, and 33.20% to 38.82%, 58.88%, 37.46%, 37.67%, and 42.24%, respectively. We detected that distinct LMWOAs had their own optimal concentrations for treating Sb-contaminated soil. Generally, the extraction efficiency increases with increasing concentration because of more reactants. For OA, TA, and MA, removal efficiencies climbed to maximum values before they achieved constant levels or fluctuated in small ranges. Nevertheless, For HEDP and CA, excessive concentration of the washing agents would, on the contrary, reduce the washing efficiencies, especially HEDP. When the HEDP concentration reached 1.0 mol/L, the Sb removal efficiency was only 35.43%, which was a 10.07% decrease compared to its maximum value at 0.2 mol/L. This effect might be brought on by the washing agent solution being more viscous due to the high concentration of HEDP or CA [61]. Viscous washing agent solutions may hamper the dissociation and mobility of ions, which, in addition, may result in adequate mixing of washing agent and soil [62]. This shows that determining the optimal concentration not only improves the washing efficiency but also saves the dosage of LMWOAs used.

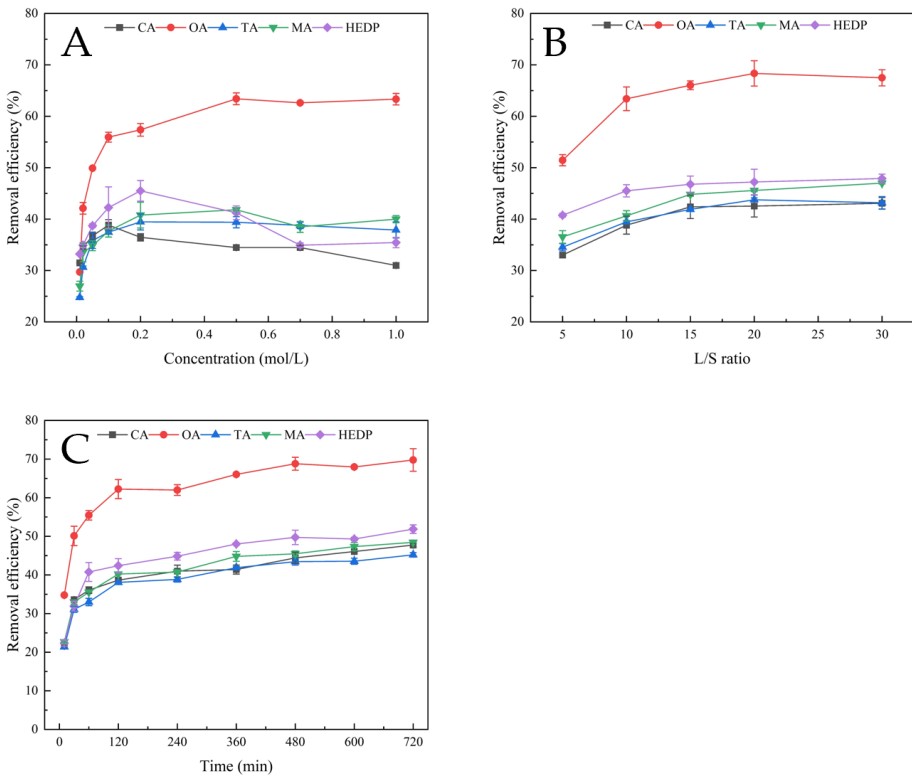

**Figure 2.** Removal efficiencies of (**A**) different concentrations of LMWOAs (washing time 360 min, L/S ratio 10/1), (**B**) different L/S ratio (optimum concentrations of LMWOAs, washing time 360 min), (**C**) different washing time (optimum concentrations of LMWOAs, L/S ratio 15/1). Other experimental conditions were: Shaking speed 250 rpm and temperature 25 °C.

Overall, the best washing agent concentration of CA was 0.1 mol/L, TA, MA, and HEDP were 0.2 mol/L, and OA was 0.5 mol/L. the best washing efficiencies of CA, OA, TA, MA, and HEDP were 38.82%, 63.40%, 39.45%, 40.79%, and 45.50%, respectively. The above optimum concentrations were used to carry out experiments for the following washing parameter.

### 3.3.2. L/S Ratio

The L/S ratio plays an essential role in soil washing and has a significant impact on both the total extraction and removal of pollutants as well as the volume of leftover wash water that needs to be dealt with [63]. Under the optimal concentration conditions, as shown in Figure 2B, there was a significant increase in washing efficiency when the L/S ratio increased from 5:1 to 10:1, and a small increase when it increased from 10:1 to 15:1. When the L/S ratio is 15:1, the washing efficiencies of CA, OA, TA, MA, HEDP for Sb were 42.37%, 66.03%, 41.88%, 44.8%, and 46.78%, respectively. However, the removal effectiveness for Sb did not vary significantly when the L/S ratio exceeded 15:1. It was challenging to release some residual Sb fixed in the interior of soil particles. As a result, the removal of Sb was only marginally enhanced by using too much mixed washing reagent [56]. A too low L/S ratio will result in inadequate contact and reaction between the soil and the drencher, while a too high L/S ratio will consume large amounts of water and energy, inevitably [64]. For those reasons, an L/S ratio of 15:1 was chosen as the optimum condition in this study. Due to these reasons, an L/S ratio of 15:1 was selected as the optimal parameter for this study.

### 3.3.3. Washing Time and Desorption Kinetic

The length of time spent washing determines the degree of heavy metal desorption, thus influencing the removal efficiency of heavy metals [65]. In this study, the washing efficiency of each LMWOA showed the same trend. As presented in Figure 2C, in the five washing reagent systems, the removal efficiency of Sb increased considerably with reaction time (0–60 min) and maintained a slow increase after 60 min. The maximum removal efficiencies were reached in about 360~480 min when the adsorption-desorption system almost reached dynamic equilibrium. For example, the removal efficiency of Sb by OA rose to 55.46% rapidly in the initial 60 min and increased from 55.46% to 66.88% gently from 60 to 480 min, and then the removal efficiency remained basically stable. HEDP removed 22.35% of Sb at 10 min, then 40.78% was removed when the washing time reached 60 min, and finally got a saturation washing rate of 49.73% at 480 min. There were two stages to the desorption of heavy metals: The quick desorption of weakly bound heavy metals from the soil surface and the gradual release of tightly bound heavy metals from the soil particles [66]. The rate-limited dissolution and desorption of heavy metals could be the main culprits behind the time-dependent washing process [67]. We suggest setting the washing time at 480 min in order to remove as much Sb as possible.

To analyze the kinetic characteristics of Sb desorption in soil, four kinetic models were selected to further describe the soil Sb removal process. The fitted results were assessed by contrasting the $R^2$ and SE. The closer R is to 1, and the closer SE is to 0, the better the fit is. As shown in Table 3, in terms of coefficients of determination, Elovich was the best kinetic model for CA, TA, and MA, and pseudo-second-order was the best kinetic model for OA and HEDP.

**Table 3.** Parameters of desorption kinetic equations of Sb.

| Extractant | Elovich $S=a+b\ln t$ | | Two-Constant $\ln S=a+b\ln t$ | | Pseudo-First-Order $\ln S=\ln S_{max}+bt$ | | Pseudo-Second-Order $1/S=a+b/t$ | |
|---|---|---|---|---|---|---|---|---|
| | $R^2$ | SE | $R^2$ | SE | $R^2$ | SE | $R^2$ | SE |
| CA | 0.933 | 0.550 | 0.903 | 0.567 | 0.779 | 0.322 | 0.922 | 0.401 |
| OA | 0.932 | 0.784 | 0.892 | 0.798 | 0.813 | 0.476 | 0.965 | 0.578 |
| TA | 0.971 | 0.539 | 0.939 | 0.558 | 0.744 | 0.331 | 0.929 | 0.399 |
| MA | 0.967 | 0.579 | 0.941 | 0.602 | 0.737 | 0.354 | 0.920 | 0.429 |
| HEDP | 0.951 | 0.670 | 0.910 | 0.689 | 0.836 | 0.469 | 0.963 | 0.520 |

$S$: Desorption content of Sb at time, $S_{max}$: Maximum desorption content of Sb, $t$: Washing time, $a$, $b$: Kinetic parameter.

Overall, it was feasible to consider both the Elovich and pseudo-second-order equations as suitable fitting models to explain the desorption of Sb by LMWOAs (Figure 3). The Elovich model was successfully employed to represent phosphorus sorption and desorption in soils [58], demonstrating that Sb release from soil may be the same as phosphorus release and may be a complicated non-homogeneous diffusion process. The two-constant kinetic equation model considers that the removal of heavy metals from contaminated soil is mainly influenced by the chemical interaction between the washing agent and the contaminant, and this model is suitable for non-homogeneous diffusion processes [49]. Results of kinetic models fitting reinforced the fact that desorption of Sb due to LMWOAs washing is a chemical reaction-driven and non-homogeneous process.

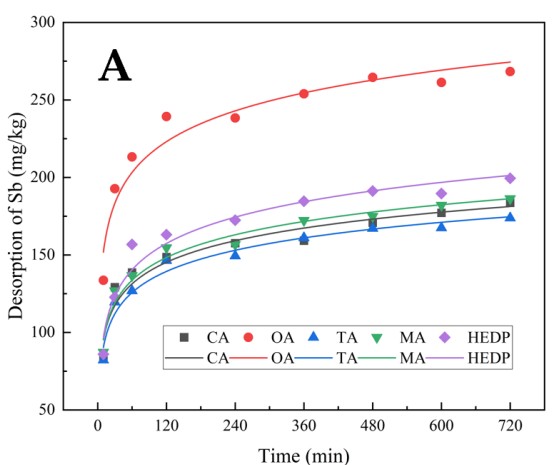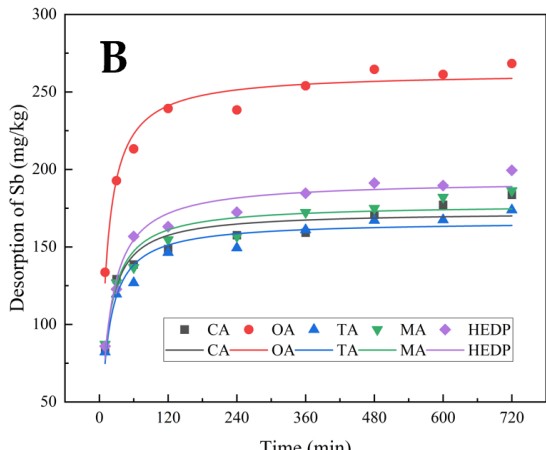

**Figure 3.** Kinetic curves of Sb washing from the soil by different agents. (**A**) Elovich equation fitting, (**B**) pseudo-second-order equation fitting.

### 3.4. Sb Fractions in the Soil before and after Washing

The biological toxicity and removal of heavy metals can be determined by the species distribution of heavy metals [68]. To investigate the migration and transformation of Sb in soils, a modified Wenzel extraction experiment was conducted. Figure 4 shows changes in Sb speciation in soil under optimal washing conditions with the first-rank concentrations of LMWOAs, 8 h washing time, and 15:1 L/S ratio. As illustrated in Figure 4A, after washing the soil samples with LMWOAs under optimum conditions, a significant amount of Sb content was removed from the soil. CA, OA, TA, MA, and HEDP removed 170.81 mg/kg (44.42%), 264.53 mg/kg (68.79%), 167.08 mg/kg (43.45%), 174.92 mg/kg (45.49%), and 191.23 mg/kg (49.73%) of Sb from the soil, respectively. Figure 4B shows the proportion of Sb in different fractions before and after washing. Sb in the original soil mainly existed in F4 (45.79%), followed by F3 (25.75%), F2 (14.97%), and F1 (13.49%). The residual fraction of Sb in soils (F4) is regarded as being geochemically immobile since it is tightly bound to the

soil matrix. F3 is considered strongly bound to Fe, Al, and Mn oxides, whose biological effectiveness and mobility are also relatively low in general environments [69,70]. The combination of F3 and F4 exceeded 70.00% of the total Sb, therefore, it could be indicated that Sb is mostly immobile in soils. In addition, previous studies have shown low levels of easy-to-migrate Sb [42].

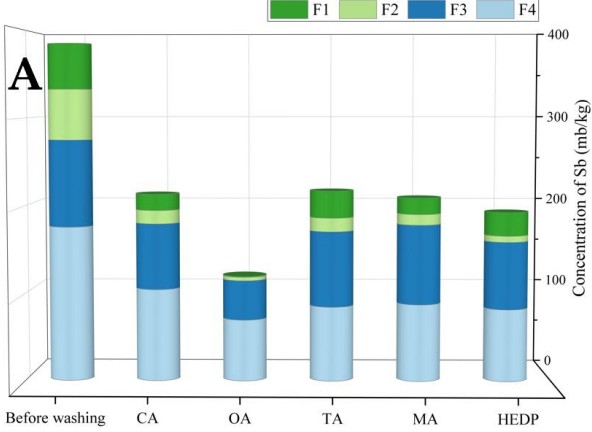
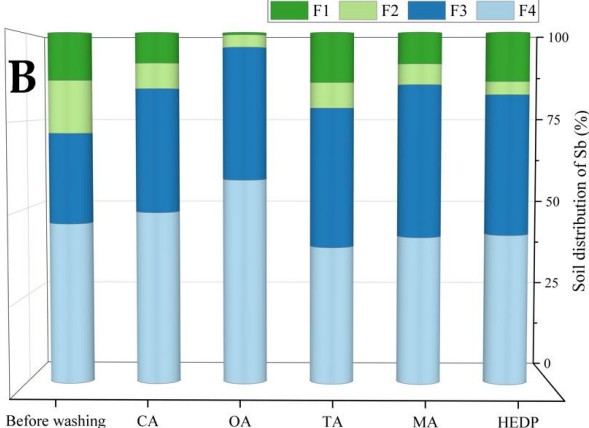

**Figure 4.** Changes in the concentration (**A**) and distribution (**B**) of Sb in different fractions. Experimental conditions were: Optimum concentrations of the LMWOAs, L/S ratio 15:1, washing time 480 min, shaking speed 250 rpm, and temperature 25 °C.

Although lowering the total amounts of heavy metals to satisfy regulations is a vital remediation goal, much more focus should be placed on the possible environmental concerns brought on by the existence of easily mobile metal components [71]. After TA and HEDP washing, the F1 proportion of soil, however, increased to 14.11% and 13.85% from 13.49%. When the soil redox conditions change because of the addition of LMWOAs, Fe/Al(III) in the soil can be reduced to Fe/Al(II), prompting the dissolution of iron or aluminum oxides, allowing Sb bound to them to enter the soil solution with enhanced mobility [72]. It is speculated that LMWOAs enter the soil and target the Fe/Al bound Sb for ligand exchange and induced dissolution to promote the conversion of bound and residual Sb to the non-specifically or specifically adsorbed state [40]. Soil washing is a complex dynamic desorption-adsorption process of heavy metals in soil, which can remove heavy metals but also change their characteristics, so the potential environmental risk of heavy metals after washing needs to be considered before application.

After OA washing, the F1, F2, F3, and F4 contents of Sb in soil decreased from 51.86, 57.57, 99.03, and 176.09 mg/kg to 0.78, 4.16, 45.04, and 70.04 mg/kg, respectively. OA is more effective than other LMWOAs. One explanation for this difference in removing Sb was that OA ($pK_{a1}$ = 1.23, $pK_{a2}$ = 4.19) added to the soil could release large amounts of $H^+$ and negatively charged organic ligands [56], which competed with oxygenated anions of Sb in the soil, facilitating desorption of Sb. The findings demonstrated that OA not only successfully decreased the concentration of total Sb but also improved the stability of Sb in soil, lowering the potential environmental risk of Sb in soil.

### 3.5. Analysis of Functional Groups in the Soil

In this study, OA possessed the highest Sb washing efficiency, with HEDP in second place. Therefore, OA and HEDP were selected as representatives of LMWOAs for FTIR analysis. The FTIR spectra of the soils before and after washing are depicted in Figure 5. The strong broadband at 3696 $cm^{-1}$ was attributed to Si–OH [73]. The peaks at 3620 $cm^{-1}$ and 3440 $cm^{-1}$ were attributed to –OH [48]. The C=C/C=O stretching vibrations were credited with producing the peaks at 1631 $cm^{-1}$ and 794 $cm^{-1}$ [48,74], 1031 $cm^{-1}$ and 1007 $cm^{-1}$ corresponded to in-plane Si/Fe–O bending vibrations [75,76]. The peak

at 912 cm$^{-1}$ was indicative of Al–Al–OH [77]. The Si–O–Si and O–Si–O of SiO$_2$ were reflected by peaks centered at 695 cm$^{-1}$ and 537 cm$^{-1}$ [76,78], and the peak of 471 cm$^{-1}$ was attributed to Si–O–Fe [39]. According to these findings, silicon, iron, and aluminum oxides make up the majority of the soil samples. In comparison with the original soils, the strength of Al–Al–OH, Si/Fe-O, and Si-O in the soils after washing was considerably weakened, especially by OA. In addition, Si–O–Si and O–Si–O of SiO$_2$ stretching vibrations shifted to a lower wavelength. It indicated that LMWOAs could facilitate the dissolution of crystalline silicon, iron, and aluminum oxides. What is more, LMWOAs can form a strong complex with Fe/Al on the surface of Fe/Al oxides, and the Fe/Al-LMWOAs complex can be detached, which causes the dissolution of the Fe/Al oxides and the consequent release of Sb associated with them [40,79].

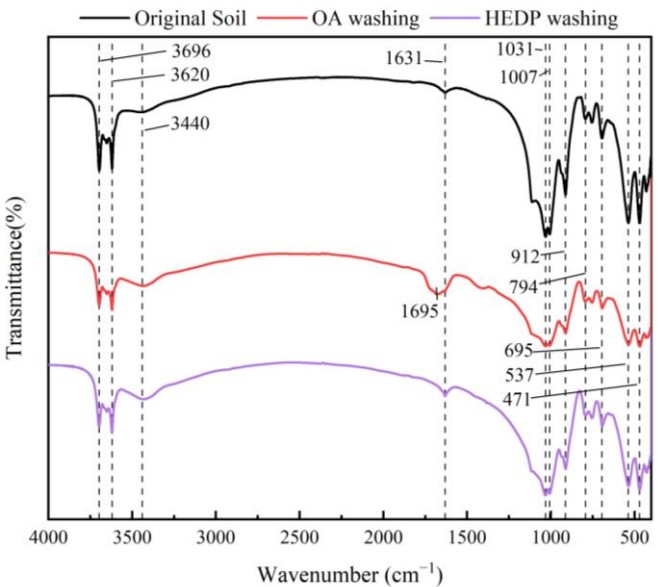

**Figure 5.** FTIR spectra of soils before and after washing with OA and HEDP.

After washing with OA, an additional peak emerged at 1695 cm$^{-1}$, indicative of the carboxyl C=O stretching vibration peak in carboxyl groups (–COOH) [27,80]. It may be due to OA residue after soil washing. Overall, washing soil with LMWOAs did not affect the soils' functional groups severely, and it was a green remediation technology with a low impact on the soils.

### 3.6. The Mineral Structure of the Soil

X-ray diffraction (XRD) analysis was applied to analyze the mineral structure of the soils before and after washing. X-ray diffraction results (Figure 6) indicated that the original soil samples were composed mainly of quartz (SiO$_2$), and kaolinite (Al$_4$(Si$_4$O$_{10}$)(OH)$_8$), muscovite (KAl$_2$(AlSi$_3$O$_{10}$)(OH)$_2$), and probably also goethite (FeO(OH)) at 2θ of 19.72° and 50.09° [12]. However, the distinctive peaks of Sb-associated crystals did not show up in the XRD patterns, which could be explained by the low Sb level and the relatively low crystallinity in the soils [80]. In summary, the soil was rich in aluminosilicate minerals and possibly even contained hydrated iron oxides, and this result was consistent with the soil being rich in iron and aluminum in Table 2. Sb is an amphoteric element mainly present in the environment as an oxygenated anion, in addition to having a significant affinity for clay and co-precipitating with aluminum, manganese, and iron [1,26,81]. However, the much higher abundance of Fe and Al oxides (hydroxide) than Mn oxides (hydroxide) in the natural environment indicates that Fe and Al minerals could govern the behavior of Sb [1,12,82]. Therefore, the aluminosilicate minerals and iron oxides in the soil were the important reasons why Sb was adsorbed by the soil.

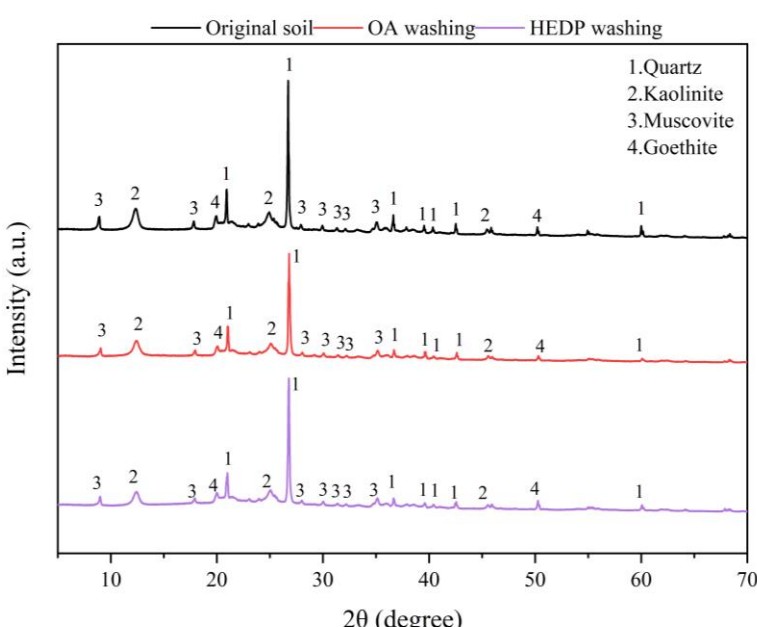

**Figure 6.** XRD patterns of soils before and after washing with OA and HEDP.

For the soils after OA and HEDP washing, peak intensities for almost all positions (2θ) fell off, except for a slight increase in the peak at 21.00°. It indicated that after washing, these minerals were partially dissolved and even transformed, which contributed to the release of heavy metals bound to these minerals from the solid phase to the liquid phase [39]. It was worth mentioning that LMWOAs removed Sb from the soil precisely by affecting the fate of Fe/Al minerals, whose mechanisms mainly included acid dissolution, complexation, and ligand exchange. Compared with the original soils, the main peaks from soils after OA and HEDP washing did not appear misaligned and were still capable of maintaining stability. Meanwhile, no significant new peaks appeared. This indicated that the effects of washing with LMWOAs, represented by OA and HEDP, on soil mineral composition and crystal structure were relatively small.

### 3.7. Soil Surface Morphology and Element Distribution

Figure 7 shows the results of an SEM-EDS analysis of the soil in order to see how the element distribution and surface morphology of the soil changed after being washed with OA and HEDP. It was visible that the surface of the original soil was very rough and had a loose and distinct layered structure. The study by Cerqueira et al. showed that foreign contaminants and soil particle surface components could form irregular crystalline bodies [83]. As an exotic contaminant, Sb was mainly bound to Fe/Al oxides in the soil, and it was hypothesized that these native minerals bound to Sb would be distributed in irregular crystals on the soil surface. After washing with OA and HEDP, the soil particle structure changed, and the surface became smoother and more compact. This may be because OA and HEDP dissolved some minerals on the soil surface and desorbed Sb from the soil surface by forming complexes. In addition, the surface morphology of the washed soil was observed with more pore structure (circled in Figure 7), a change that can be attributed to the hydraulic flushing or the dissolution of soluble salts [48,84].

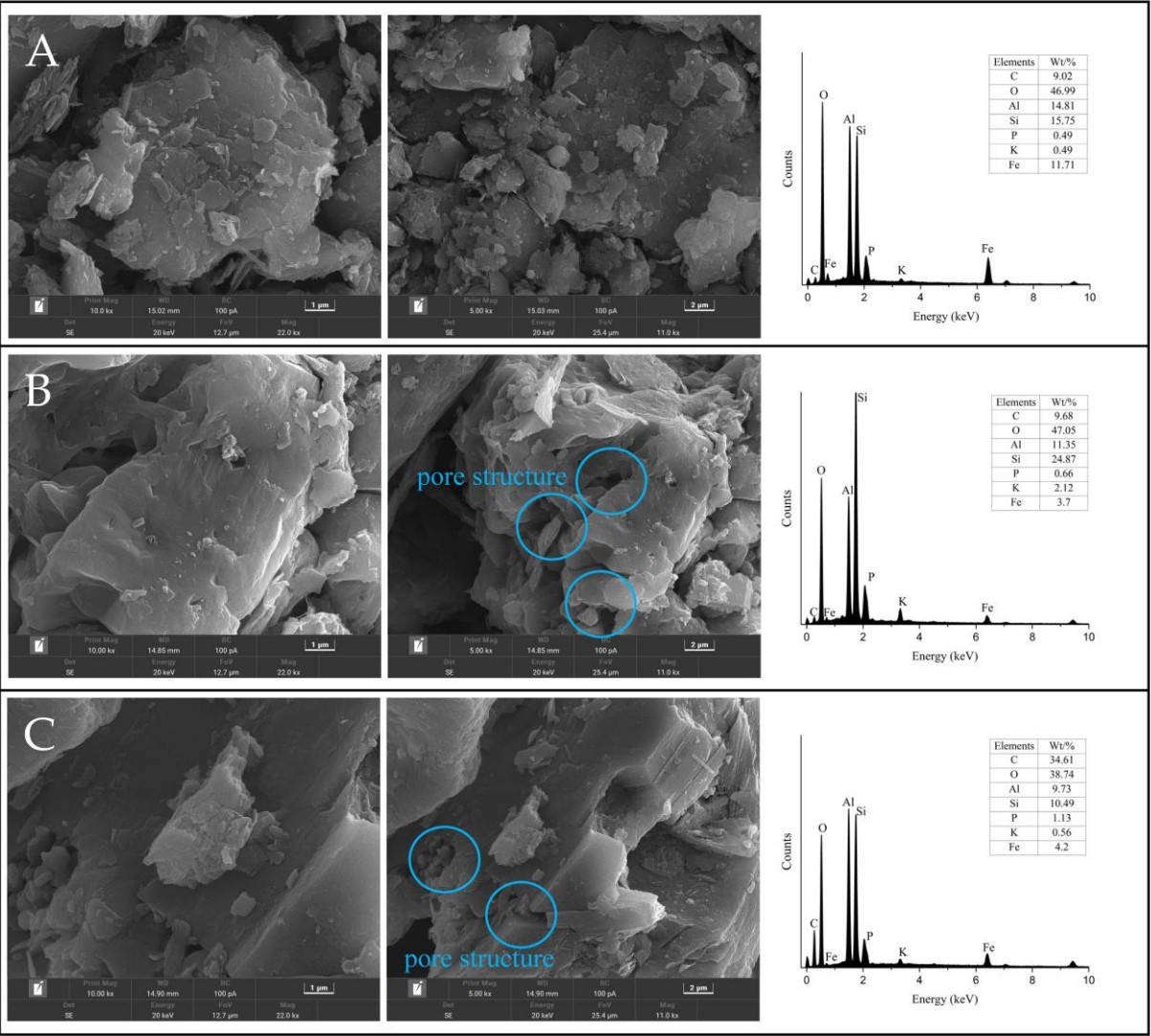

**Figure 7.** SEM-EDS images of the Sb contaminated soil before and after washing. (**A**) Original soil, (**B**) after washing with OA, (**C**) after washing with HEDP.

The EDS data further indicated that the elemental distribution of the soil has changed. Al and Fe were partially washed out of the soil by OA and HEDP washing. Combining the results of sequential extraction and XRD analysis, it could be assumed that LMWOAs remove Sb by removing amorphous or crystalline Fe and Al from the soil.

### 3.8. Potential Mechanism of Sb Removal

Based on the above results, the potential mechanisms of Sb removal by LMWOAs washing were summarized as follows (Figure 8): (1) Acid dissolution. LMWOAs provide $H^+$, which keeps Sb-contaminated soil in an acidic environment and promotes the dissolution of soil minerals containing contaminants. The minerals here include mainly iron and aluminum oxides and hydroxides. In addition, Sb adsorbed on the soil surface is released due to the dissolution, (2) complexation: The groups such as carboxyl or hydroxyl groups of LMWOAs can complex with soil humus or heavy metals like Fe/Al and form water-soluble complexes [25], which release oxygenated anions of Sb originally complexed with, (3) ligand exchange: Negatively charged organic ligands or phosphate groups of HEDP can compete with the oxygenated anions of Sb in the soil for adsorption sites and undergo ligand exchange.

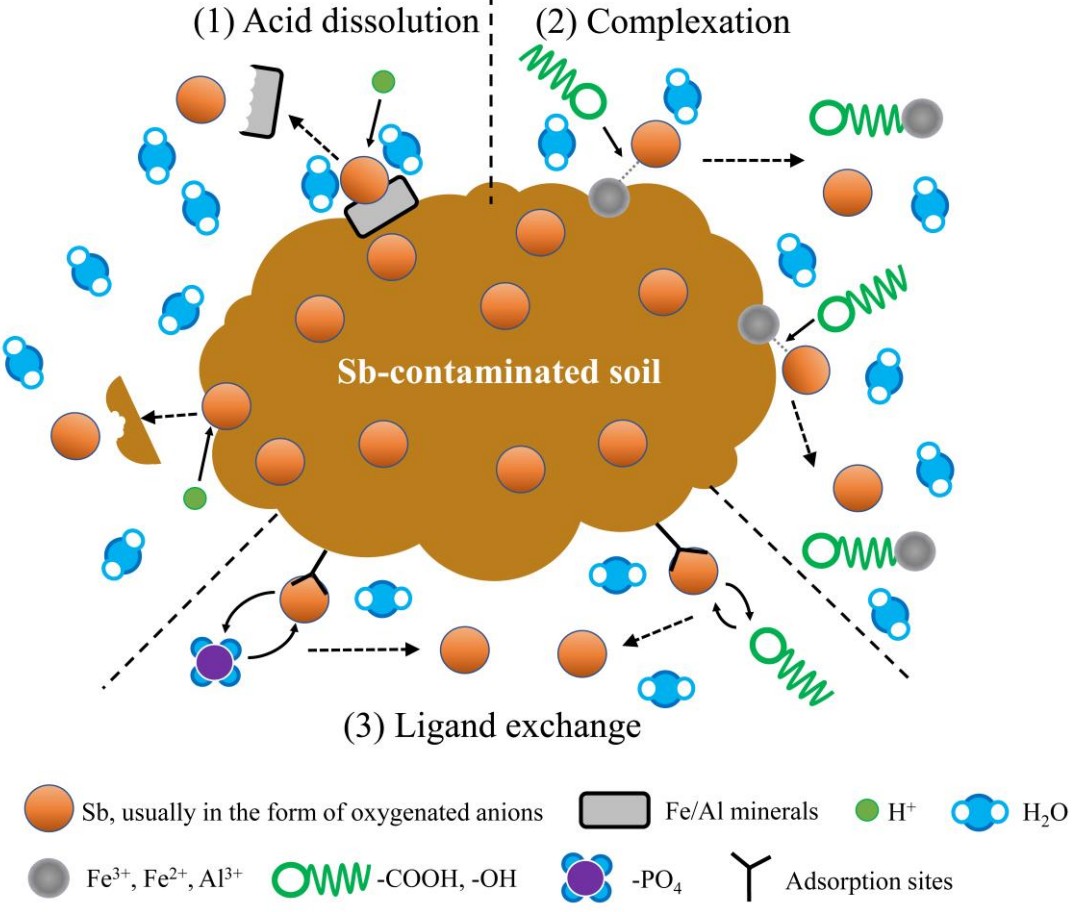

**Figure 8.** Schematic of the potential mechanisms of Sb removal from soil by LMWOAs washing.

## 4. Conclusions

The efficiencies and mechanisms of Sb removal from soil using LMWOAs washing were studied. The results demonstrated that the optimal washing agents were 0.5 mol/L OA, 0.2 mol/L HEDP, 0.2 mol/L MA, 0.1 mol/L CA and 0.2 mol/L TA, respectively. The optimal washing parameters were identified as an L/S ratio of 15:1 and a washing time of 480 min. The removal efficiencies at these parameters were: OA (68.79%) > HEDP (49.73%) > MA (45.49%) > CA (44.42%) > TA (43.45%). The sequential extraction results showed that the Sb in the soil existed mainly in the residue state and the Fe/Al oxides bound state, and among the five LMWOAs, OA removed the largest amount of easily mobile components of Sb. According to the kinetic experiments, the desorption of Sb due to LMWOAs washing was a chemical reaction-driven and non-homogeneous process. Both Elovich and pseudo-second-order equations could be considered satisfactory fitting models to describe the desorption of Sb by LMWOAs. Sb was mainly bound to Fe/Al minerals in the soil, and LMWOAs removed Sb from the soil precisely by affecting the fate of Fe/Al minerals. Therefore, the elution of Sb was accompanied by the loss of Fe/Al. We concluded that the main mechanisms of removal of Sb by LMWOAs were acid dissolution, complexation, and ligand exchange. In addition, LMWOAs did not significantly change the mineral phase and functional groups in the soil, hence, soil washing with LMWOAs was a green remediation technique.

Overall, this work has shown that soil washing with LMWOAs, especially OA and HEDP, can be a green and effective technology for the remediation of Sb-contaminated soil. Moreover, the potential mechanisms of Sb removal are initially discussed in this paper in order to provide a reference for other studies on the remediation of Sb-contaminated soil.

The obtained results can provide practical information and technical support for actual soil-washing technology for the remediation of Sb-contaminated soil.

**Author Contributions:** Conceptualization W.P., S.L. and L.T.; methodology, S.L.; software, S.L.; validation, S.L., Y.Z. and Y.H.; formal analysis, S.L.; investigation, S.L. and Y.H.; resources, W.P.; data curation, S.L.; writing—original draft preparation, S.L.; writing—review and editing, S.L., L.T. and W.P.; visualization, S.L. and X.H.; supervision, W.P.; project administration, W.P.; funding acquisition, W.P. All authors have read and agreed to the published version of the manuscript.

**Funding:** This research received no external funding.

**Institutional Review Board Statement:** Not applicable.

**Informed Consent Statement:** Not applicable.

**Data Availability Statement:** The data in this study is available from the corresponding author on reasonable request.

**Conflicts of Interest:** The authors declare no conflict of interest.

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
