# Peer review of "Remediation of Sb-Contaminated Soil by Low Molecular Weight Organic Acids Washing: Efficiencies and Mechanisms"

_sustainability, doi:10.3390/su15054147_

Round 1

Reviewer 1 Report

The article entitled Remediation of Sb-contaminated soil by low molecular weight organic acids washing: efficiencies and mechanisms presents some interesting and good results. However, the below-mentioned major revision remarks must be addressed before the second round of revision.

1. Rewriting this Abstract is necessary: It ought to be succinct and provide readers with information on the background, research question, hypothesis, methodology, key findings, and conclusions of the study that is being presented. Ideally, it should also discuss the main implications and wider context of your findings.

2.   The methodology needs to be strengthened. It should be clear and logical so that repeating your work will be possible for interested researchers. If the methodology, or a portion of it, has already been published elsewhere, you should give a brief summary and cite the source.

 3. The introduction section requires a more detailed discussion leading to this study's problem statement and scope. Also, more literature is needed to be discussed. Lines 41-43 please support this statement by citing https://doi.org/10.1016/j.scitotenv.2022.159213; in lines 49-51 please cite: https://doi.org/10.1016/j.scitotenv.2023.161468

4. The novelty of your work should be apparent and additionally highlighted, together with the objectives of your research, in the last paragraph of the Introduction. Moreover in line 

5. Section 3.6: Mineralogy and geotechnical characteristics of native soil are not well highlighted. Discussion is required on the role of soil mineralogy in relation to Sb-contamination and washing agents.

7. Lines 206-208 need revision; please make sure grammatical and formatting mistakes do not appear in your manuscript. Also, Line 169-172 no need of mentioning the software name instead cite references such as https://doi.org/10.1007/s11356-021-16912-w. Check the formatting of your manuscript in detail. Further "soil particles after washing showed cracks" in Line 441 is nontechnical; the native soil used by authors is more of showing clay fabric thus crack can not be a good technical word to describe porous structure formed by the washing and Sb contamination. Please revise this line accordingly and to support the porous variation in SEM please highlight changes on the SEM Figures.

8. The conclusion should provide all the novel and important findings of this study. Please revise it. Moreover, the significance of your work needs to be stated more explicitly.

Reviewer 2 Report

The results of the study is quite sufficient and revealed quite a lot of informations on this particular area, though a thorough and more comphrehensive discussion referring to other studies is also  important in order to be abble to justify our findings

Thus, the discussion part can be improved

1.     The study was meant to seek the most effective washing methods for cleaning up antimon (Sb) contaminated soils by using organic acids which are believed to be relatively more effective and environmental friendly

2.     I believe the topic is original since limited publication on such study are found and is relevant in the field, although still need further research to achieve a novelty and an applicable  method

3.     There are some add informations, includes the effective concentrations and effective time duration for certain organic acids in washing the contaminated soil and some advanced methods

4.     The specific improvement  in analysis methods such as using such a complete laboratory equipments in one research set, Sequential extraction of Sb from soil as a modified Wenzel extraction method was used to sequentially extract different forms of Sb from the soil.

5.     Yes, the main questions and hypothesis are presented in the conclusions

6.     Yes, appropriate, the quantity and the recency

7.     The graphs are clearly and very well presented, the axis, the units etc. eazy to be understood, communicative. The SEM-EDS images are quite clear. Paticularly figure 8 can be considered as a new valuable information.

Round 2

Reviewer 1 Report

Paper can be accepted in it's current form